# Mo Contact via High-Power Impulse Magnetron Sputtering on Polyimide Substrate

**Yung-Lin Chen** [1], **Yi-Cheng Lin** [1,*] and **Wan-Yu Wu** [2]

1   Department of Mechatronics Engineering, National Changhua University of Education, Changhua 50074, Taiwan; windcloudalex9@gmail.com
2   Department of Materials Science and Engineering, Da-Yeh University, Dacun, Changhua 515006, Taiwan; wywu@mail.dyu.edu.tw
*   Correspondence: ielinyc@cc.ncue.edu.tw; Tel.: +886-4-7232105; Fax: +886-4-7211149

**Abstract:** It has always been a huge challenge to prepare the Mo back contact of inorganic compound thin film solar cells (e.g., CIGS, CZTS, $Sb_2Se_3$) with good conductivity and adhesion at the same time. High-power impulse magnetron sputtering (HiPIMS) has been proposed as one solution to improve the properties of the thin film. In this study, the HiPIMS technology replaced the traditional DC power sputtering technology to deposit Mo back contact on polyimide (PI) substrates by adjusting the experimental parameters of HiPIMS, including working pressure and pulse DC bias. When the Mo back contact is prepared under a working pressure of 5 mTorr and bias voltage of $-20$ V, the conductivity of the Mo back contact is $9.9 \times 10^{-6}$ Ω·cm, the residual stress of 720 MPa, and the film still has good adhesion. Under the minimum radius of curvature of 10 mm, the resistivity change rate of Mo back contact does not increase by more than 15% regardless of the 1680 h or 1500 bending cycle tests, and the Mo film still has good adhesion in appearance. Experimental results show that, compared with traditional DC sputtering, HiPIMS coating technology has better conductivity and adhesion at the same time, and is especially suitable for PI substrates.

**Keywords:** HiPIMS; Mo back contact; PI flexible; CIGS





## 1. Introduction

The advantages of flexible thin film solar cells are their light weight and flexibility [1–3]. Li [4] examined the characteristics of different flexible substrates, among which the glass transition temperature ($T_g$), high thermal stability, light weight, and transparency of PI substrates make them the most suitable substrate for flexible inorganic compound thin film solar cells (e.g., CIGS, CZTS, $Sb_2Se_3$) [5,6]. As is well known in CIGS or Sb2Se3 solar cell, Mo film is used for the back contact, because it has low resistivity, CTE compatibility, high stability during the selenization process, and good ohmic contact with CIGS absorber [7,8]. DC sputtering is used to deposit the Mo back contacts of conventional CIGS, CZTS, and $Sb_2Se_3$ solar cells; however, Mo back contacts deposited using DC sputtering adhere to plastic substrates poorly and have more defects, often resulting in overly high residual stress that causes peeling in the absorber layer during high-temperature thermal annealing. Thus, reducing residual stress in Mo films is crucial [9,10]. Vink [7] mentioned that tuning the working pressure for the coating can produce different patterns and magnitudes of stress and alter the microstructures of Mo films. To lower the residual stress in Mo film and enhance film adhesion, Lin [11] adopted a double-layered Mo film structure in which higher working pressure was applied to deposit the bottom layer to improve adhesion and lower working pressure was used to deposit the top layer to reduce the resistivity. The resistivity of the resulting Mo film was $34 \times 10^{-6}$ Ω·cm, and the residual stress was 430 MPa. Although the residual stress of the Mo film was controlled and its adhesion increased, this fabrication method was more complex and produced poorer resistivity.

Recently, a method of HiPIMS was proposed as a solution to improve the properties of thin films [12–15]. Lundin [16] pointed out that high-power impulse magnetron sputtering (HiPIMS) can substantially enhance the ionization rate of the plasma and is a better option for depositing high-quality thin films than DC sputtering. Studies [17,18] have also mentioned that the structure of the thin film can vary with the sputtering technique and coating conditions. Furthermore, applying pulse DC bias to sputtering processes [19–22] can effectively reduce film roughness and produce thin films with better crystallization. Bobzin [19] explained that increases in the bias voltage increase the kinetic energy of the incident particles and effectively enhance the crystallization of the thin film but that an overly high bias voltage can lead to re-sputtering.

Most studies [23,24] indicated that the mechanical properties of flexible solar cells are very important; Lee et al. [25] bent the solar cells 1000 times to demonstrate good stability of the electrode. This study used two bending tests to verify the mechanical reliability of the Mo film on a flexible PI substrate: (1) retention testing and (2) cycle testing. Overall, this study employed the new HiPIMS coating technique to deposit a Mo film on a PI substrate for the back contacts of CIGS and $Sb_2Se_3$ solar cells. We experimented with different coating conditions, including working pressure and pulse DC bias, in hopes of improving the properties of Mo back contacts deposited using DC sputtering, achieving lower resistivity and good reliability by controlling the residual stress in the Mo film. We realize a single layer structure of the Mo film to simplify the complex fabrication process of the Mo film with the double-layer structure.

## 2. Materials and Methods

A molybdenum (99.99% and six-inch) sputtering target was employed, and Mo back contacts were deposited on $2 \times 2$ cm$^2$ PI plastic substrates using HiPIMS (Huettinger Highpulse 4001 G2, Ditzingen, Germany) in current control mode with sputtering power of 1.5 kW, Ar flow rate of 30 sccm, frequency of 250 Hz, pulse width of 80 µs, and duty cycle of 2% as in Figure 1. We also created a control group with a molybdenum (99.99%) sputtering target and conventional DC sputtering (Advanced Converter PMP-1, Denver, CO, USA) at sputtering power of 200 W and working pressure of 5 mTorr [11,26]. The thickness of the two types of Mo back contacts was fixed at $780 \pm 50$ nm. We first experimented with the following working pressures for HiPIMS coating: 3, 5, and 7 mTorr. This allowed us to identify the working pressure with the best results in terms of resistivity, structural properties, and residual stress of the Mo back contact film. In the second phase, we applied the following pulse synchronized substrate bias (Huettinger Bias 4010 G2, Ditzingen, Germany): floating, $-20$, $-40$, and $-60$ V. The aim was to synchronize the pulse-on time of the Mo ions generated by HiPIMS with the $T_{on}$ time of the pulse bias applied to the substrate and minimize the excessive bombardment of the Ar and Mo particles on the thin film. This further improved the resistivity and structural properties of the Mo back contact film. We also confirmed the influence of the morphology and residual stress of the thin film. Finally, we employed two bending tests, as in Figure 2, to verify the reliability of using HiPIMS to deposit Mo back contacts on Pi substrates for flexible CIGS solar cells. In the cycle testing, we manually bend the sample, and stop bending every 500 cycles. We measure resistivity every period, and continue the bending test until 1500 cycles.

The morphology and grain growth of the Mo film were observed using a field-emission scanning electron microscope (FE-SEM, JEOL JSM-7800F Prime, Tokyo, Japan) operating at 3 kV with a 50,000 magnification rate. To observe the surface roughness of the Mo film, we used a scanning probe microscope (Solver P47-PRO, Moscow, Russia) in semi-contact mode. We took measures at five sites ($5 \mu m \times 5 \mu m$), removed the best and worst values, calculated the average of the three other values, and expressed the result using the root mean square (RMS). We then used an X-ray diffractometer (Bruker D8 discover, Billerica, MA, USA) and the symmetric Bragg diffraction approach to analyze the crystal structures of the Mo film (CuK$\alpha$ radiation with wavelength of $\lambda = 1.54052$ Å, measured in the θ-2θ scan mode at a speed of $2°/$min). We can measure the residual stress in the film resulting

from different incident angles, tilt angles, Poisson's ratios, and Young's modulus. Finally, we used a Hall measurement system (ECOPIA HMS-2000 manual, Anyang, South Korea) to measure the resistivity and carrier mobility of the film.

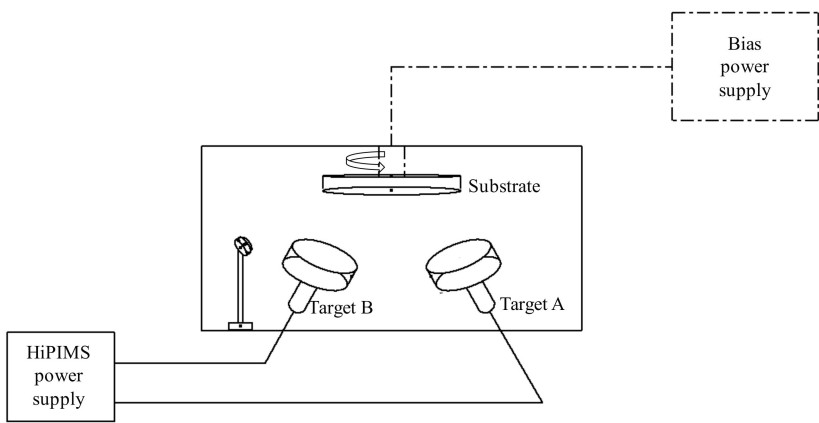

**Figure 1.** Schematic diagram of HiPIMS sputtering system.

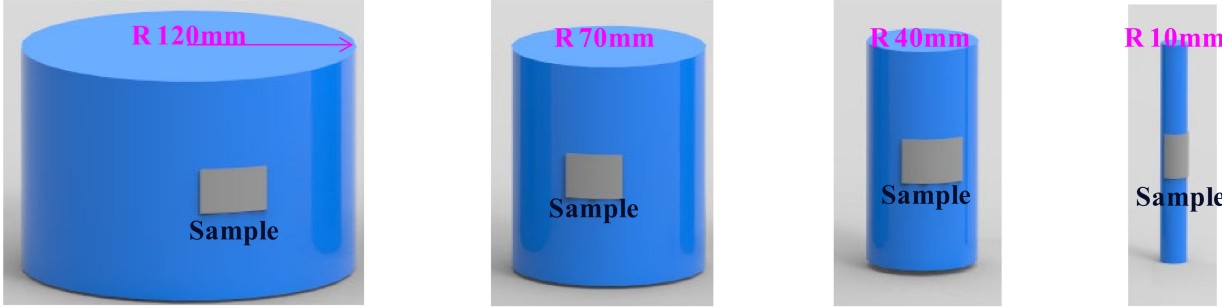

**Figure 2.** Schematic of Mo films bent to different curvature radii.

## 3. Results and Discussion

### 3.1. Influence of HiPIMS Parameters on Mo Structure Properties

To understand the differences in Mo film morphology resulting from HiPIMS with different working pressures and conventional DC sputtering, we observed the SEM images in Figure 3. The film surface resulting from DC sputtering in Figure 3a presents fine crystallite scales, whereas the Mo films resulting from HiPIMS with working pressures 3 mTorr and 5 mTorr show visible fish-like grains on the surface. The grains are dense and have distinct boundaries. However, as the working pressure increases to 7 mTorr, the fish-like grains gradually disappear and lose their distinct boundaries. According to microstructure zone theory regarding the physical vapor deposition of thin films [27], the forms of crystal grains vary with substrate temperature and the kinetic energy of deposited atoms. Due to the process characteristics of HiPIMS, incident particles may have high kinetic energy, thereby producing high-density thin films. The Mo film was thus denser and had nanocrystalline grains. In contrast, high working pressure greatly reduces the kinetic energy of deposition, creating long strip-shaped crystalline surface microstructures that are not as dense. These results are similar to those of the aforementioned structure zone model [27].

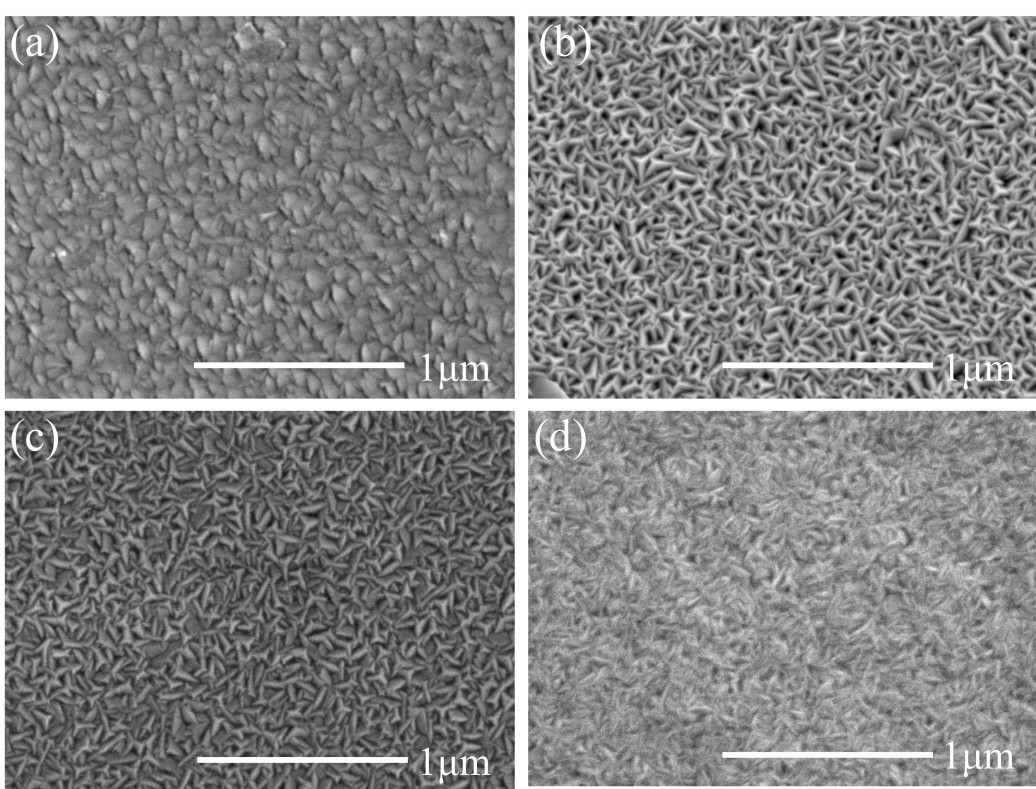

**Figure 3.** The SEM morphology of Mo film resulting from HiPIMS with different working pressures and conventional DC sputtering: (**a**) DC sputtering; (**b**) 3 mTorr; (**c**) 5 mTorr; (**d**) 7 mTorr.

　　　Figure 4 displays surface roughness of the Mo films resulting from different working pressures, observed using AFM. As the working pressure was gradually increased from 3 mTorr to 7 mTorr, the RMS of surface roughness increased from 3.8 nm to 6.1 nm. These films deposited using low working pressure presented denser fibrous structures, creating relatively flat surface morphology. However, the RMS values showed no significant differences and remained below 10 nm, which is relatively flat and conducive to the subsequent stacking of the absorber layer, as mentioned by Vink [7].

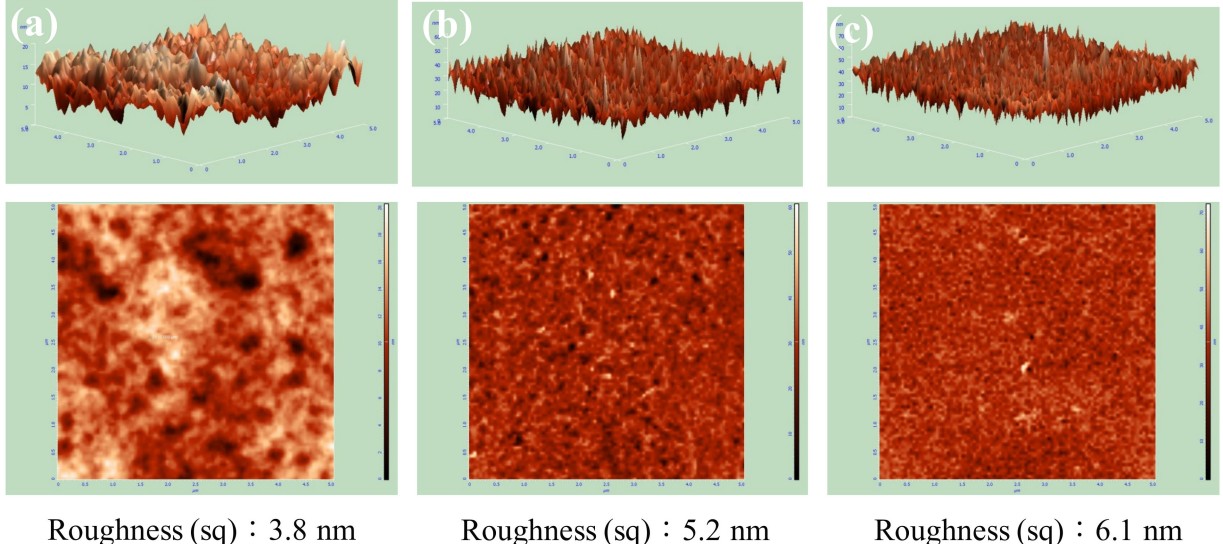

Roughness (sq)：3.8 nm　　　　Roughness (sq)：5.2 nm　　　　Roughness (sq)：6.1 nm

**Figure 4.** Surface roughness of the Mo films resulting from different working pressures: (**a**) 3 mTorr; (**b**) 5 mTorr; (**c**) 7 mTorr.

To understand the influence of conventional DC sputtering and HiPIMS with different working pressures on the crystal structures and residual stress of Mo film, we conducted XRD analysis for confirmation, the results of which are exhibited in Figures 5 and 6. In Figure 5, the main peaks identified at around $2\theta = 40.5°$ and $73.7°$ from (110) and (211) planes, respectively. After confirming JCPDS (Joint Committee on Powder Diffraction Standards), it is the BCC (body-centered cubic) structure. At the lower working pressure of 3 mTorr, the crystallographic planes in the Mo film were along the (110) orientation. Under low working pressure, the mean free path of sputtered ions increases, which means that they are less likely to collide with gas molecules and dissociate. As a result, the energy of ions reaching the substrate increases and leads to a better crystallinity of the Mo film. This also leads to better resistivity and clear fish-like structure, which we explain in the results of the electrical property analysis and SEM analysis, respectively. This result is consistent with the observations made by Jubault [28]. In addition, changing working pressure also leads to the shifts of diffraction peaks. With DC sputtering and HiPIMS under low working pressure, the diffraction angles shift more significantly toward smaller angles.

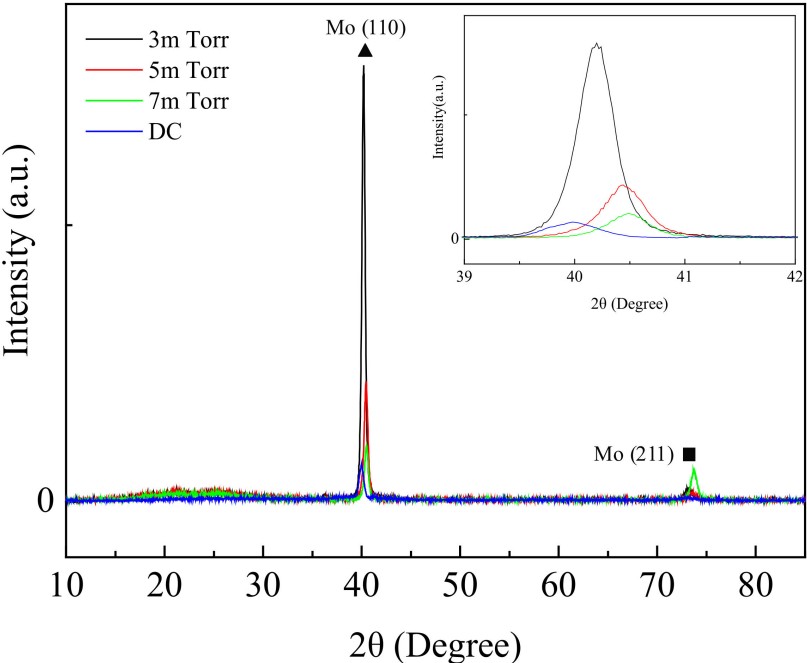

**Figure 5.** XRD analysis of conventional DC sputtering and HiPIMS with different working pressures of Mo film.

As shown in Figure 6, the displacement of XRD diffraction peaks is used to analyze the residual stress generated by different working pressures. It is clear that greater residual stress was produced in the films fabricated using conventional DC sputtering and HiPIMS at 3 mTorr, which were 5178 MPa and 4004 MPa respectively. A much lower residual stress was obtained as the working pressures are at 5 and 7 mTorr. This is due to a higher working pressure leading to a shorter mean free path and significantly reducing the adatom energy and the residual stress of the resulting films. The residual stress was also observed in the XRD peak shift. The $2\theta$ diffraction angle of (110) at $40.5°$ shifted towards a smaller angle, especially for DC sputtering and 3 mTorr HiPIMS, indicating a changed lattice distance and distorted crystal structures. The structural stability and reliability of flexible components are crucial, and overly high residual stress may alter the lattice constant and affect the mechanical properties. This, in turn, decreases the stability and reliability of the Mo film. It can also promote blister-induced peeling during the thermal processing of CIGS films. The working pressure should therefore be optimally set at 5 mTorr to prevent overly high residual stress from affecting the stability and reliability of component structures.

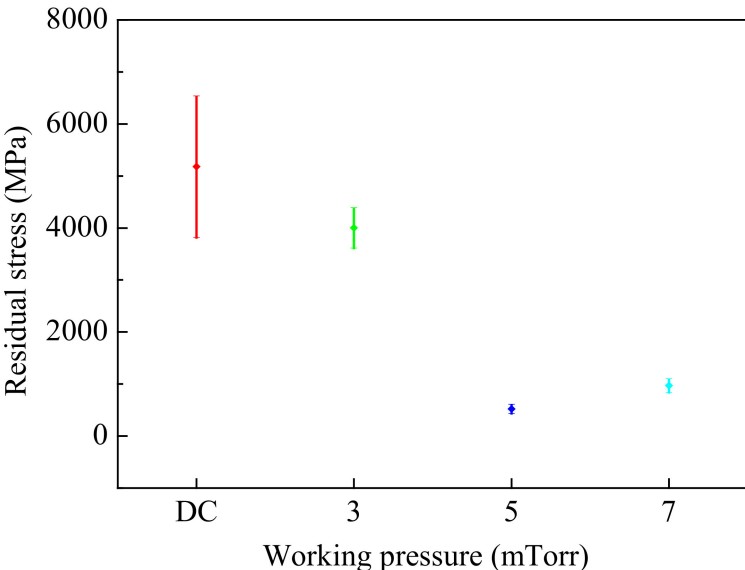

**Figure 6.** Applied XRD to analyze the residual stress resulting from different working pressures.

According to the analysis of working pressure, fixing HiPIMS working pressure at 5 mTorr is optimal. We employed SEM to observe the surface morphology and cross-section of Mo films fabricated using HiPIMS with different bias voltage, as shown in Figure 7. Figure 7a displays the Mo film surface with floating bias applied. This resulted in denser fish-like grains and a smaller and more uniform grain size. When the bias voltage was increased to −20 V, the fish-like grains on the surface became less dense, the porous structures gradually disappeared, and larger grains began to appear, as shown in Figure 7b. This was due to the connecting and merging of smaller grains, which was a result of higher particle deposition kinetic energy from the bias voltage. At the same time, it promoted the diffusion and movement of surface particles. Messier [29] mentioned that as the bias voltage increases, film structures become denser and crystallization improves. This inference corresponds with results of the subsequent electrical property analysis. In Figure 7c, the bias voltage was increased to −40 V, and thin elongated column-shaped crystals began to appear. When the bias voltage was increased to −60 V, distinct column-shaped crystals appeared, and they were larger and irregular. The fish-like structures were also gone by this time. Therefore, moderately increasing the kinetic energy of incident particles can effectively promote film growth and enhance density and crystallization. However, overly high kinetic energy in the incident particles can alter the orientation of the crystals and enable the particles to bombard the film surface, which will create defects. As for the cross-sections, there are dense amorphous structures on the Mo film surfaces when floating bias was applied. When the bias voltage was increased to −20V, larger crystals could be observed. When the bias voltage was increased to −40 V, column-shaped crystals began to appear. When the bias voltage reached −60 V, the Mo film exhibited column-shaped crystalline structures, but in irregular directions, as shown in Figure 7d, which is not conducive to carrier transmission.

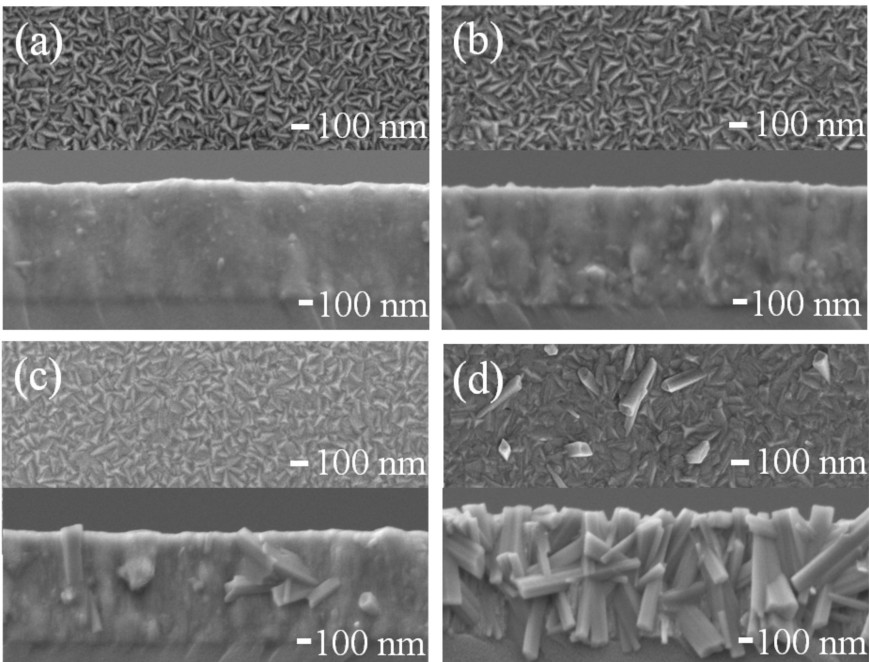

**Figure 7.** SEM to observe the surface morphology and cross-section of Mo films fabricated using HiPIMS with different bias voltage: (**a**) floating; (**b**) −20 V; (**c**) −40 V; (**d**) −60 V.

Figure 8 displays the AFM-measured surface roughness of the Mo films fabricated under different bias voltage. Regarding the deposition of ITO films, Lee et al. [30] indicated that the surface roughness of ITO films increases with the substrate bias. Our results are consistent with this trend. As the bias voltage was increased from floating to −20 V, the RMS of the Mo film surface roughness rose from 5.2 nm to 12.9 nm. Next, when the bias voltage was increased to −40 V to −60 V, the roughness increased significantly to 34.3 nm and 39.8 nm, respectively. In the subsequent solar cell component stacking, Mo film surfaces that are uneven and overly rough have an adverse impact on the adhesion of the absorber layer and cause leakage currents, which reduce the efficiency of components and affect component stacking.

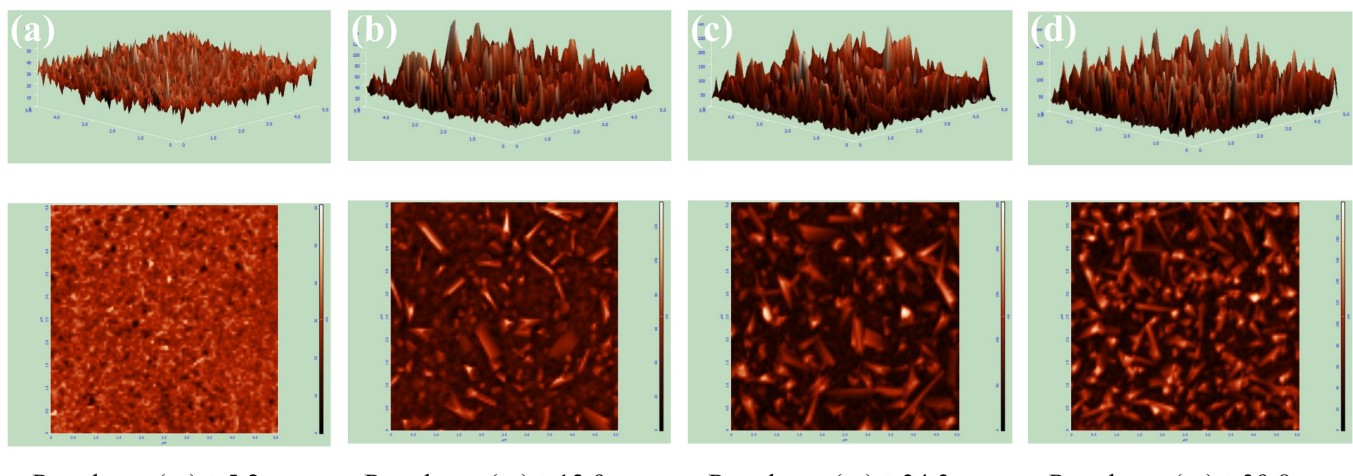

| Roughness (sq)：5.2 nm | Roughness (sq)：12.9 nm | Roughness (sq)：34.3 nm | Roughness (sq)：39.8 nm |

**Figure 8.** Surface roughness of the Mo films fabricated under different HiPIMS bias voltage (**a**) floating; (**b**) −20 V; (**c**) −40 V; (**d**) −60 V.

Since an overly high bias voltage may lead to re-sputtering during the deposition and damage the film structure [19], a pulsed DC bias was applied at a fixed working pressure of 5 mTorr in this study. Figure 9 shows the XRD results of Mo films obtained at various pulsed DC bias. As can be seen, when floating bias was applied, the intensity of the main diffraction peak of the Mo film mainly had a (110) peak, seconded by the (211) peak of higher diffraction angles. When the bias voltage was −20 V, the intensity of the diffraction peak with the (110) peak increased significantly, thereby representing better crystal quality and clear fish-like structure in the Mo film. When the bias voltage continued to increase from −40 V to −60 V, the intensity of the (110) peak declined, whereas the (211) diffraction intensity of the higher diffraction angles increased. This is because the overly high kinetic energy of the incident particles made it easier for the particles to form crystals, but the direction was mainly (211): there tended to be more defects, and because the direction of the crystals was irregular, the surface roughness of the film increased.

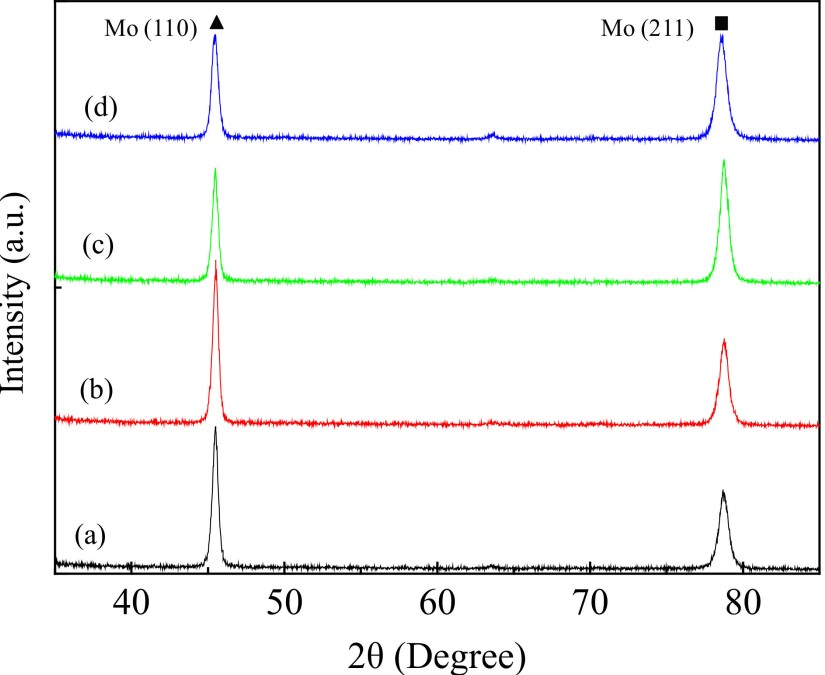

**Figure 9.** XRD analysis of HiPIMS with different bias voltage of Mo film: (**a**) floating; (**b**) −20 V; (**c**) −40 V; (**d**) −60 V.

To prevent overly high incident carrier energy from accumulating residual stress in the Mo film, we used XRD analysis to measure the residual stress resulting from different pulse biases, as shown in Figure 10. The results show that the residual stress in the Mo films resulting from different pulse DC biases increases with the negative bias. When the bias voltage was −20 V, the residual stress in the film was 720 MPa and did not increase significantly. When the bias voltage continued to increase to −40 V and −60 V, the residual stress rose to 1950 MPa and 1700 MPa, respectively. This result is consistent with what was previously described. Residual stress affects the structural stability and reliability of components.

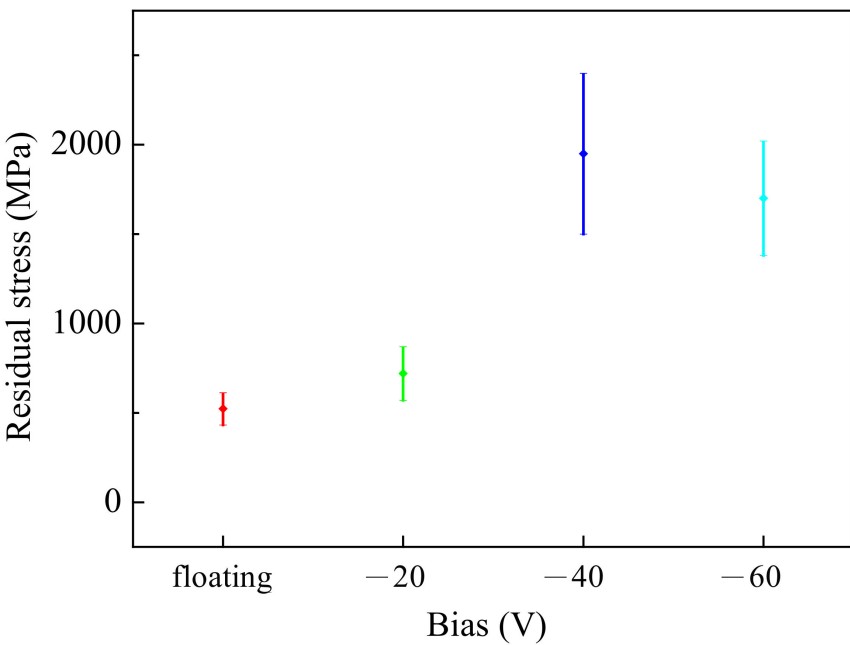

**Figure 10.** XRD analysis to measure the residual stress resulting from different bias voltage.

### 3.2. Influence of HiPIMS Parameters on Mo Electrical Properties

Figure 11 compares the electrical properties, measured using a Hall measurement system, of the Mo films fabricated using HiPIMS with different working pressures and conventional DC sputtering. As can be seen, the resistivity of the Mo film resulting from DC sputtering was $45.5 \times 10^{-6}$ Ω·cm, and the carrier mobility was 13.6 cm$^2$/Vs. Using the lower HiPIMS working pressures of 3 mTorr and 5 mTorr, excellent resistivity of $12.6 \times 10^{-6}$ Ω·cm and $18.3 \times 10^{-6}$ Ω·cm could be respectively obtained. Due to the increased carrier mobility, these results are significantly better than those derived using the DC sputtering conditions [31]. This is because the high coating power density and dissociation rate of HiPIMS make the structural characteristics and surface defects of the Mo films better than those resulting from DC sputtering and facilitate carrier movement. Furthermore, the mean free paths of the molecules are longer, thereby enabling them to have greater deposition kinetic energy and produce film structures with better nucleation and crystallization, which is consistent with the contents of the XRD analysis. In contrast, the mean free paths of the molecules during Mo film deposition using the higher working pressure of 7 mTorr were shorter, entailing a higher chance of molecule collision, poorer film crystal quality, and increased structural defects—such as poor density, dislocation, and grain boundaries. These characteristics greatly reduce carrier mobility and significantly increase resistivity. These results are similar to those derived in a previous study [32].

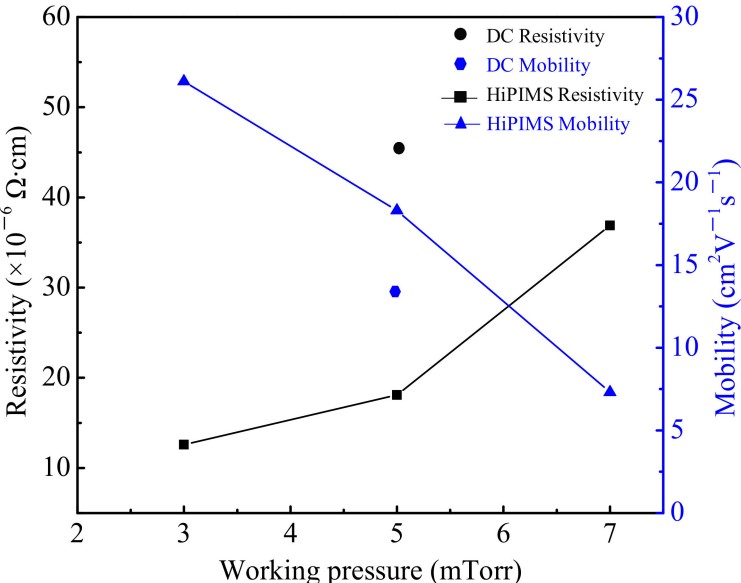

**Figure 11.** Electrical properties of the Mo films fabricated using HiPIMS with different working pressures and conventional DC sputtering.

Figure 12 presents the influence of HiPIMS at a fixed working pressure of 5 mTorr and different bias voltages on the electrical properties. The figure shows that when the bias voltage was increased to −20 V, carrier mobility increased to 39.5 cm$^2$/Vs. The resistivity improved significantly, decreasing to $9.95 \times 10^{-6}$ Ω·cm. The increase in carrier mobility can be attributed to the improved crystallization of the Mo film. This is because the application of a suitable pulse bias can further enhance the kinetic energy of incident particles, giving the Mo film greater density and better crystallization during deposition, reducing the losses caused by free electrons at the grain boundaries or defects in the crystal structures, and effectively increasing carrier mobility. Increasing the bias voltage to −60 V produced the best carrier mobility at 54.8 cm$^2$/Vs, but the resistivity only improved slightly, decreasing to $9.01 \times 10^{-6}$ Ω·cm. Overly high bias voltage causes ions to bombard the substrate at high speed and also promotes re-sputtering, which is not conducive to film growth [19,33].

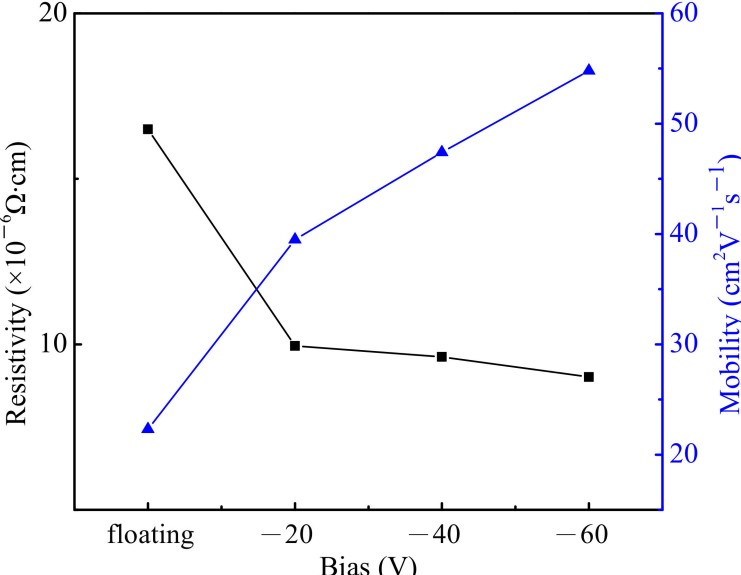

**Figure 12.** Influence of HiPIMS at different bias voltages on the electrical properties.

*3.3. Reliability Tests of the HiPIMS Deposited Mo*

To understand the stability of the Mo film on PI substrates, we bent different Mo film samples fabricated using the same coating conditions (working pressure of 5 mTorr and bias voltage of −20 V) to the following curvature radii: R10, R40, R70, and R120 mm. The results of resistivity change rate (R.C.R) are displayed in Figure 13a. As can be seen, the resistivity change rate of the sample bent to the smallest curvature radius (i.e., R10 mm) showed a slight increase from 0% to 12.5% after 1680 h (70 days), changing around 12.5%. We speculate that, under prolonged stress, minute cracks formed in the Mo film, causing the resistivity to increase slightly. In contrast, the resistivity of the Mo films bent to the other curvature radii displayed no significant changes. These results demonstrate that Mo films fabricated with working pressure of 5 mTorr and bias voltage of −20 V in HiPIMS have enhanced reliability and durability in applications with different curvature radii. The analysis results in Figure 13b show the resistivity change rate of the Mo films, following subjection to different numbers of bending cycles. This analysis helped us understand the durability of the Mo film on a PI substrate. We found that after the samples were bent 1000 times, the resistivity change rate rose slightly. After 1500 times, the resistivity change rate of the Mo film increased from 0% to 12.7%, the changes remaining around 10% to 15%, which confirms the trends found in previous studies.

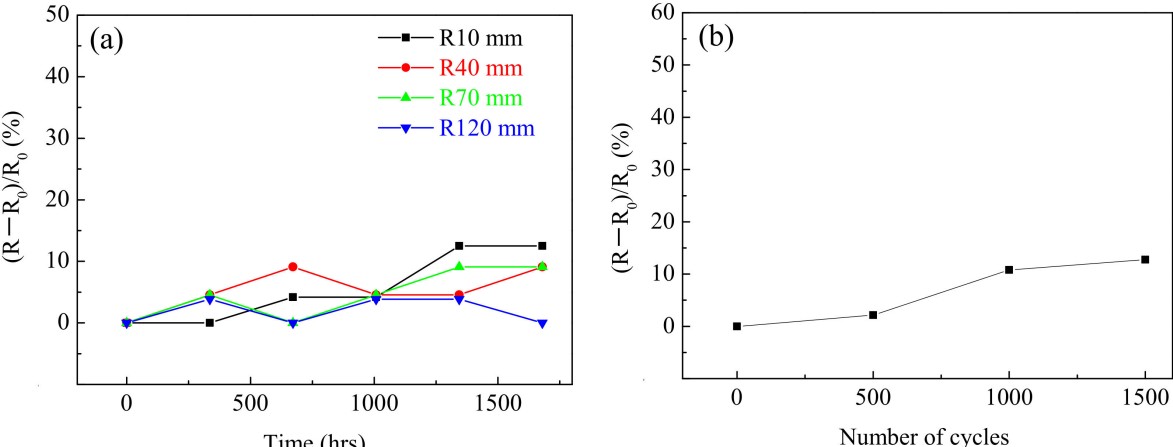

**Figure 13.** Resistivity change rate of the mechanical reliability test: (**a**) retention testing and (**b**) cycle testing.

## 4. Conclusions

This study employed HiPIMS to deposit Mo films on flexible PI plastic substrates for the back contacts of inorganic compound thin film solar cells (e.g., CIGS, CZTS, and $Sb_2Se_3$). We examined the influence of HiPIMS parameters on the structural and electrical properties of the Mo film and identified the optimal parameters and reliability. The results indicated that Mo film deposited under working pressure of 5 mTorr produces lower resistivity at $18.3 \times 10^{-6}$ Ω·cm and lower residual stress at 520 MPa, both of which are better than those resulting from conventional DC sputtering. Under these conditions, applying suitable bias voltage of −20 V can increase the intensity of the Mo(110) orientation, promote better crystallization, effectively reduce the resistivity to $9.95 \times 10^{-6}$ Ω·cm, result in a flat surface with an RMS roughness of 12.9 nm, and produce lower residual stress in the film at 720 MPa. Finally, bending and reliability tests (both retention and cycle tests) presented only slight changes in the resistivity change rate (10% to 15%), thereby demonstrating that HiPIMS can reduce the residual stress in the Mo film, enhance its adhesion, and reduce resistivity. This enables the replacement of complex double-layered Mo film structure with a single Mo film layer.

**Author Contributions:** Y.-L.C. is the main author; methodology, Y.-L.C.; validation, Y.-C.L. and W.-Y.W.; formal analysis, Y.-C.L.; writing—original draft preparation, Y.-L.C.; writing—review and editing, Y.-C.L. and W.-Y.W.; supervision, Y.-C.L.; project administration, Y.-L.C. All authors have read and agreed to the published version of the manuscript.

**Funding:** This research received no external funding.

**Institutional Review Board Statement:** Not applicable.

**Informed Consent Statement:** Not applicable.

**Data Availability Statement:** Not applicable.

**Conflicts of Interest:** The authors declare no conflict of interest.

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
