# Peer review of "Mo Contact via High-Power Impulse Magnetron Sputtering on Polyimide Substrate"

_coatings, doi:10.3390/coatings12010096_

Round 1

Reviewer 1 Report

This manuscript describes the deposition of molybdenum back contacts for solar cell devices using HiPIMS. DC bias and deposition pressure are used as the variables, SEM, AFM, XRD and Hall effect measurements are reported.

The paper details how the measurements were made and how the analysis was performed.  Lines 87 to 99 detail the methods.  Figures 4 and 8 have error bars.  This is good.
The conclusions are founded on the data presented.  There are no unfounded or questionable claims in the conclusions.
A suggestion: the abstract could be shortened.  Start with line 23 "Experimental.." And then add line 18 to 23, "When the Mo.."

Well done.  Publish as is.

Author Response

The answers to reviewers' comments has been resubmitted as the attachment. We look forward to your positive response.

Reviewer 2 Report

The article of Chen et al. presentes interesting results concerning the use of HiPIMS technique for growth Mo thin films on polyimide substrate. The use of Mo on polyimide is for thin film solar cells application (back contact) and one of the most frequent problem is the adhesion of Mo on polyimide surface. In general, the article is well-written and the results relevant for the research field. It is recommended its publication after some improvements: 

(1) Abstract: Please define PI substrate when this was cited for the first time in abstract.

(2) Introduction: Please inform at the last paragraph of the introduction topic the novelty of the present work with relation to literature;

(3) Materials and Methods: Please inform the manufacturer and model of the HiPIMS, DC and bias power supplies. Also, inform the dimension of the Mo sputtering target. Suggestion: present a schematic diagram of the HiPIMS apparatus;

(4) Materials and methods: Please correct Figure "A". Also, improve de description of Figure 3, only "Results" is not a description. Correct the number sequence of the Figures along the article;

(5) Please inform the section "3. Results and discussion" after materials section.

(6) The title of Figs 11a and 11b should be merged into a single title.

Author Response

(The authors gave the same response as above.)

Reviewer 3 Report

The manuscript entitled : “Mo Contact via High-power Impulse Magnetron Sputtering on Polyimide Substrate” describe the study of the Mo metal layer deposited on flexible polymer and testing their capability for bending.

At first, they motivate the purpose of the work to try to suit the special issue. I think the paper is not appropriate for the special issue, but the work or study worth to be taken place. The authors need to work more on the introduction to motivate the publication of this work in this special issue. For example, why Mo as electrode. There are other materials we would think for electrodes ….

There are many issues in this manuscript and inconsistency that really needs to be corrected or answered before a possibility to be considered for publications.

My main concerns are :

The XRD:

  • A bcc has its most intense peak 110 and second most intense 211 etc . So, the authors should reconsider their description and explanation on XRD results. The main orientation is not 110 but it is just the natural most intense peak observed on a randomly oriented bcc.
  • Measurement questionable:
    • Measurement done on a HRXRD, but the steps are big (0.03 degree per sec) not detail on time/steps, how was the alignment done, is the height correct? , on some you see a bump around 20, corresponding to the substrate. Why is the DC which seems to be well crystallized on SEM is looking almost with low intensity , is the alignment correct?
    • A height difference will generate a shift of the peaks !!!
    • How is the stress measured?
      • HiPIMS films are usually having a lot of stress compared to DC : please check the literature
      • What is the reference not stressed sample is ?
      • I would recommend other method for measuring the stress such as the sin”Psi2” or the multy hkl stress measurement. Still using XRD. I guess the nominal values are off due to this…
    • “To prevent overly-high incident carrier energy from accumulating residual stress in the Mo film”. I do not understand this sentence (the wording might be wrong.
  • Deposition of the film:
    • no pulse bias means floating (if so, what is the value) is it grounded)
    • very hard to just compare 3 HiPIMS done with different pressure with one DC at one pressure …. Especially discussing and drawing conclusion from different pressure HiPIMS deposited with the DC
      • Have the authors considered removing the DC sample from the manuscript
    • Electrical measurement
      • The bending was done with the sample place on a mold .
        • Why do the authors show the sheet resistance and not the resistivity, you just need to multiply by the thickness?
        • What is the sheet resistance before bending (reference) as this is made using same condition, but the sheet resistance could be different on different sample?
        • “The increase in carrier mobility can be attributed to the improved crystallization of the Mo film” isn’t it more due to the change or morphology? Please rethink the discussion part.
        • The authors mentioned bending the sample several times upt to 1500 . Have they done it by hand that many times and also how and where were place the probes for measuring the sheet resistance?

Author Response

(The authors gave the same response as above.)

Round 2

Reviewer 2 Report

The authors made corrections to the article, which is why its publication in its current form is recommended.

Author Response

Thanks for your suggestion. The revised manuscript has been resubmitted to your journal. We look forward to your positive response. But, we are not very clear about this comment.  This comment is the abstract we have submitted in original manuscript.  We have revised the abstract according to Reviewer#1's in the first revision.

Reviewer 3 Report

The authors did improve the manuscript after the several recommendations from the reviewers.

The manuscript is overall hard to follow. At first, I think it is due to the English and flow which can be revised, I suggest the editorial board to have a look. Here are few examples:

  • Wording: Work pressure (mTorr): working pressure (which in figure 11 refer to the pressure and “DC” (not worrect)
  • Bobzin [19] mentioned that overly-high DC bias may lead to re-sputtering during film deposition, which damages the structure of the film. We therefore fixed the working pressure at 5 mTorr, …. .
  • To understand why the lattice distortion mentioned above causes the diffraction peak values to shift, we applied XRD to analyze the residual stress resulting from different working pressures
  • At the same time, angle shifts in the diffraction peaks of the Mo film can be observed as the working pressure changes.
  • substrate have more energy, thereby producing better thin film structures and diffraction peak values. (better diffraction peak values?)

Those are few examples. In the meantime, the authors do not necessarily describe well the figure before discussing it.

Some other minor things to correct:

>Experimental part.

- The author mentioned that the condition for the DC deposited film was taken from optimized condition from references. No references is mentioned precisely in the experimental part.

- The figure 1 is in my opinion not necessary, It does not bring more information than a educational picture of the chamber.

- The bending test was not clear and still not in the manuscript. The authors did describe the process well in the answer to reviewer and should include it in the manuscript. It is bending the sample, many times manually and then measuring the resistivity with the sample flat.

- Authors should use the resistivity to be coherent all along the manuscript, not once the sheet resistance and then the resistivity.

- The method to measure the stress is named the sin”psi”2 (https://ora.ox.ac.uk/objects/uuid:003ab986-3be1-48ac-ace0-f430627aec7f) and when applying the authors should mention which poison ratio there are using.

>results

- The authors mentioned in the answer to reviewer that they measure the density of the sample by XRR. The authors could potentially compare the value between sample  (maybe even to the theoretical value) and determined the apparent density of the material ("porosity"). Having such values will help discussing the density of the film with numbers.

- “At the lower working pressure of 3 mTorr, the crystallographic planes in the Mo film was along the (110) c-axis preferred orientation, which was perpendicular to the substrate. The diffraction peak at (110) peak of the Mo film presented the highest intensity.”

wording wrong :

  • “Mo film was along the (110) c-axis preferred”, if it is (110) it can’t be c-axis
  • the 110 peak of Mo is the highest for a randomly oriented film. what the authors mean is the relative ratio of intensities between the peak differs than for a randomly oriented film. To be honest I would recommend to not emphasis too much on this. Telling that there might be a certain degree of orientation of grains differing between the sample but stay low . This does not change the paper story and does not lead to over interpreting things with maybe the wrong wording.

- “It is clear that greater residual stress was produced in the films fabricated using conventional DC sputtering and Hi- PIMS with lower working pressure, which was 5,178 MPa and 4,004 MPa respectively.”

It is not that clear as if you check the error bar, there are the same. . within the Hi-PIMS sample, yes there are two very low.

One aspect that the author did not comment, is why are those films low stressed? . Most of film deposited by HiPIMS present high level of stress (GPa).

- When the authors describe the part on the bias study: “We employed SEM to observe the surface morphology and cross-section of Mo films fabricated using HiPIMS with different pulse bias”

It is confusing. As in the experimental part they mentioned once that the bias is pulsed and synchronised. I would recommend to use “different bias voltage” , and when there is no bias applied the authors should write “floating”.

- “The peak angle of the original (110) diffraction angle 2. of 40.5 ° shifted towards a smaller angle, indicating that the atoms in the Mo structure are not well-arranged.”

Very hard to understand, By definition having a peak of diffraction means a well-arranged atoms array. So please rethink that part.

- Title of 3.3 “Influence of HiPIMS Parameters on Mo Reliability Tests”

It is not the influence of the HiPIMS parameters studied there. There is only one sample (or one condition) . It is simply mechanical bending test on the best sample.

Author Response

The revised manuscript has been resubmitted to your journal. We look forward to your positive response.
